# Influence of the Topology of Poly(L-Cysteine) on the Self-Assembly, Encapsulation and Release Profile of Doxorubicin on Dual-Responsive Hybrid Polypeptides

**DOI:** 10.3390/pharmaceutics15030790

**Published:** 2023-02-27

**Authors:** Dimitra Stavroulaki, Iro Kyroglou, Dimitrios Skourtis, Varvara Athanasiou, Pandora Thimi, Sosanna Sofianopoulou, Diana Kazaryan, Panagiota G. Fragouli, Andromahi Labrianidou, Konstantinos Dimas, Georgios Patias, David M. Haddleton, Hermis Iatrou

**Affiliations:** 1Industrial Chemistry Laboratory, Department of Chemistry, National and Kapodistrian University of Athens, Panepistimiopolis, Zografou, GR-15771 Athens, Greece; 2Hellenic Police Headquarters, Forensic Science Division, Chemical and Physical Examinations Department, GR-10442 Athens, Greece; 3DIDPE, Dyeing, Finishing, Dyestuffs and Advanced Polymers Laboratory, University of West Attica, 250 Thevon Street, GR-12241 Athens, Greece; 4Laboratory of Pharmacology, Faculty of Medicine, University of Thessaly, Viopolis, GR-41500 Larissa, Greece; 5Department of Chemistry, University of Warwick, Gibbet Hill, Coventry CV4 7AL, UK

**Keywords:** polypeptides, drug delivery, nanoparticles, doxorubicin, cancer, topology of poly-l-cystein

## Abstract

Τhe synthesis of a series of novel hybrid block copolypeptides based on poly(ethylene oxide) (PEO), poly(l-histidine) (PHis) and poly(l-cysteine) (PCys) is presented. The synthesis of the terpolymers was achieved through a ring-opening polymerization (ROP) of the corresponding protected *N*-carboxy anhydrides of *N^im^*-*Trityl*-l-histidine and *S*-*tert*-butyl-l-cysteine, using an end-amine-functionalized poly(ethylene oxide) (*m*PEO-NH_2_) as macroinitiator, followed by the deprotection of the polypeptidic blocks. The topology of PCys was either the middle block, the end block or was randomly distributed along the PHis chain. These amphiphilic hybrid copolypeptides assemble in aqueous media to form micellar structures, comprised of an outer hydrophilic corona of PEO chains, and a pH- and redox-responsive hydrophobic layer based on PHis and PCys. Due to the presence of the thiol groups of PCys, a crosslinking process was achieved further stabilizing the nanoparticles (NPs) formed. Dynamic light scattering (DLS), static light scattering (SLS) and transmission electron microscopy (TEM) were utilized to obtain the structure of the NPs. Moreover, the pH and redox responsiveness in the presence of the reductive tripeptide of glutathione (GSH) was investigated at the empty as well as the loaded NPs. The ability of the synthesized polymers to mimic natural proteins was examined by Circular Dichroism (CD), while the study of zeta potential revealed the “stealth” properties of NPs. The anticancer drug doxorubicin (DOX) was efficiently encapsulated in the hydrophobic core of the nanostructures and released under pH and redox conditions that simulate the healthy and cancer tissue environment. It was found that the topology of PCys significantly altered the structure as well as the release profile of the NPs. Finally, in vitro cytotoxicity assay of the DOX-loaded NPs against three different breast cancer cell lines showed that the nanocarriers exhibited similar or slightly better activity as compared to the free drug, rendering these novel NPs very promising materials for drug delivery applications.

## 1. Introduction

Cancer remains one of the leading causes of death worldwide. Due to its complex nature, multiple metabolic pathways and ability to resist numerous drugs, so far, selective elimination of cancer cells without influencing healthy tissues has not yet been achieved [1,2].

Most of the anticancer drugs used are poorly water soluble, leading to poor absorption and low bioavailability. Therefore, it is necessary to design and synthesize smart drug delivery systems which can transport therapeutic agents in a timely and spatially controlled manner. This need can be satisfied by using nanotechnology for the synthesis of nanostructured materials as drug carriers, the most common being micelles [3,4,5], liposomes [6], polymersomes, magnetic NPs and mesoporous silica NPs as well as hydrogels [7,8,9]. Among the materials used, polymers play a pivotal role due to their high functionalization. Drugs can be encapsulated or chemically conjugated to polymers, lowering their toxicity and increasing their solubility as well as their circulation time in the blood and preventing renal clearance for more efficient accumulation in the solid tumor either through the EPR effect or an active targeting mechanism [10].

Ideally, smart drug delivery systems to target cancer cells should have the ability to bypass numerous biological obstacles such as vasculature and non-vasculature barriers and tumor microenvironment as well as intracellular barriers [11]. The tumor microenvironment is dynamic and is characterized by acidity, hypoxia and ischemia. Usually, the pH value of a tumor tissue is 6.3–6.8, while higher concentrations of various biological substances are detected in cancer cells such as GSH and matrix metalloproteinase 2 (MMP2) as well as active oxidative species (ROS) [12,13]. The blood serum has a pH value of 7.4, while the endosomal and extracellular pH of the cancer tissues exhibits a pH range from 6.3–6.8, and that of the lysosomal compartments of the cell can be between pH 5.5–5.0 [14,15]. Therefore, pH-responsive systems can induce controlled drug release and penetrate deeper into cancer cells, due to pH variations between the intracellular organelles of the cell and the extracellular matrix. In the case of polypeptides bearing an amine or carboxylic acid in their side chains, pH changes activate the protonation/deprotonation mechanism, leading to the disassembly of their conformation and the desired triggered release of their cargo.

Therefore, the NPs that will be used to treat cancer should exhibit the following main characteristics: good biocompatibility and minimal cytotoxicity without systemic side effects, highly selective accumulation in pathological tissues, accurate stimulus responses to result in selective release of the cargo and long-term stability in blood circulation as well as minimal cargo loss before arrival at the target. In addition, drug delivery systems should be inert and stable in the aggressive environment of the blood compartment. We envisioned synthetic NPs that fulfill the above properties and can change their structure when they reach the pathological environment, releasing the cargo in a controlled way, leading to efficient and highly selective elimination of cancer cells.

PHis is a unique polypeptide with the ability to respond to physiological pH variations as its imidazole ring with a pKa around 6.5 can be protonated and deprotonated within the physiological values, altering the hydrophilicity of the polypeptide [16]. Bilalis et al. [17] have described the synthesis of PHis-grafted mesoporous silica NPs (MSNs) which can efficiently encapsulate the anticancer drug DOX, and can release it in a pH-controlled manner. Our group has also referred to the synthesis of linear and 3-miktoarm star hybrid polypeptides based on PHis, poly(l-glutamic acid), poly(l-lysine) and PEO. These novel structures could load the anticancer drug Everolimus and release it in response to pH changes [18].

Redox responsive nano-structures, whose action is determined by the microenvironment of cancer tissue, also induce controlled drug release [19,20]. The redox reaction depends on the concentration of active substances in the cell organelles. For example, the concentration of GSH is four times higher in cancerous tumors compared to its concentration in healthy tissues. Different concentrations of GSH were also found in intracellular (~2–10 mM) and extracellular (~2–10 μM) fractions. In particular, intracellular compartments comprising the cytoplasm, mitochondria, and nucleus contain higher concentrations of GSH compared to extracellular fluids [21,22,23]. It is well known that disulfide bonds are sensitive to GSH, as GSH can cause the rapid cleavage of disulfide bonds, leading to cytosolic delivery of anticancer agents [24,25,26,27].

Wang et al. [28] managed to synthesize redox-responsive SCL micelles based on poly(ethylene glycol)-*b*-poly(l-cysteine)-*b*-poly(l-phenylalanine) triblock copolymers, which could load DOX. A sustained release profile of DOX was observed from these NPs where the PCys was in a block form. Moreover, Wu et al. [29,30] reported the synthesis of DOX-loaded and gold-embedded micelles based on poly(l-cysteine) which exhibit synergistic chemo- and photothermal therapy of cancer cells. In both works, PCys was in a block form and a similar sustained release of the drug was observed. In a previous publication by our group [31], we presented the synthesis of disulfide crosslinked polypeptide nanogels consisting of PHis and PCys, which show satisfying pH, redox and thermo-responsiveness to the external stimuli. In that work, the disulfide bonds were randomly distributed along the polymeric chains. We have shown that the macromolecular architecture and topology of the blocks can play a critical role on the self-assembly of amphiphilic polymers [32,33,34,35,36]. Although the topology of cysteine along the polypeptide chain can play a critical role at the release profile of the drug, to our knowledge, there is no publication so far which studies this parameter on the structure and drug release profile of NPs. 

Herein, we present the synthesis of three series of hybrid polypeptide copolymers composed of PEO, PCys and PHis. PEO was always the first block in all these series. At the first series, PCys was the second block, while PHis was the last block. At the second series, PHis was the second block while PCys was the last. Finally, at the third series, PHis and PCys were randomly distributed along the chain. Two different compositions of PHis and PCys were used, keeping the total number of the monomeric units of PHis and PCys the same, while the PEO block was the same for all hybrid polypeptides. It was found that the empty hybrid terpolymers can self-assemble into micellar nanostructures and exhibit pH and redox responsiveness. Moreover, they can effectively load the anticancer drug DOX in differently structured NPs compared to the empty one and can release in a controlled manner, in response to pH and redox variations. The release profiles depended on the structure of the NPs. In order to elucidate the influence of PCys topology at the release profile of DOX, the release was studied in media with various concentrations of GSH as well as different pH values. Finally, in vitro studies of the efficacy of the NPs in breast cancer cell lines prove that the DOX-loaded NPs could be potentially used for cancer treatment. In order to elucidate the impact of the PCys topology on encapsulation efficiency and release profiles, the results were compared to the NPs obtained under similar conditions with similar polymers such as PEO-*b*-PHis or PEO-*b*-Poly(sarcosine)-*b*-PCys (PEO-PSAR-PCys).

## 2. Experiment

### 2.1. Materials and Methods

#### 2.1.1. Materials

Ethyl acetate (>99.9%, Carlo Erba, Val de Reuil, France) was fractionally distilled over phosphorous pentoxide. Tetrahydrofuran (THF) (>99.9%, Carlo Erba) was purified over Na-K alloy, using standard high vacuum techniques. The purification of *N*,*N*-Dimethylformamide (DMF) (99.9%, Alfa Aesar, Waltham, MA, USA anhydrous, amine free) was performed by short-path fractional distillation under high vacuum in a custom-made apparatus, and the middle fraction was used. The final product was stored in a vacuum flask at 3 °C. Benzene (99%, thiophen-free grade, Sigma Aldrich, Saint Louis, MO, USA) was treated with calcium hydride and was allowed to be stirred overnight for moisture removal. It was then distilled under high vacuum and stored in a different flask containing *n*-BuLi. Diethyl ether (>99.5%), Dichloromethane (99.8%) and Chloroform (>99.8%) were purchased from Fluka, Charlotte, NC, USA. *n*-Hexane (>95%) was obtained from Carlo Erba, Val de Reuil, France. Methyl Sulfoxide (DMSO) (99.8+%, for peptide synthesis) was supplied from Acros Organics, Waltham, MA, USA. BOC-His(*Trt*)-OH (>99%) was acquired from Christof Senn Laboratories AG, Dielsdorf, Switzerland. Sarcosine (98%) was purchased from Alfa Aesar. *S*-tert-Butylmercapto-l-Cysteine (99%) was provided by Sigma Aldrich. Methoxypolyethylene glycol amine (*m*PEO-NH_2_) with average *M*_n_ = 10,000 g mol^−1^ was obtained from Sigma Aldrich. Triethylamine (Et_3_N) (99.83%, Fluka) was dried over calcium hydride for one day and then distilled and stored under vacuum over sodium. Triphosgene (99%), Thionyl chloride (99.5+%), Hydrogen peroxide (ACS reagent, 30 wt.%, solution in water, non-stabilized) and DL-1,4-Dithiothreitol (99%) were provided by Acros Organics, Waltham, MA, USA. (R)-(+)-Limonene (97%) was purchased from Alfa Aesar. Trifluoroacetic acid (TFA) (>99%) was obtained from Fluka. Triisopropylsilane (98%) was supplied from Sigma Aldrich. l-Glutathione (98%, reduced form) was obtained from FluoroChem, Hadfield, UK. Sodium Chloride (99.9%) was purchased by Penta chemicals, Prague, Check Republic. Sodium Hydroxide pearls (99.4%) was acquired by Lachner, Neratovice, Check Republic. Hydrochloric Acid 1 mol L^−1^ and Acetic Acid glacial (99.8%) were obtained from Chem-Lab, Zedelgem, Belgium. Tris base ultrapure (99.9%) was purchased from Duchefa Biochemie, RV Haarlem, Netherlands. Sodium Phosphate Monobasic (98–100.5%) was supplied from Riedel-de Haen, Charlotte, NC, USA. Doxorubicin Hydrochloride (>99%) was obtained from Selleckchem, Planegg, Germany. Distilled water was further purified by a Milli-Q Direct Water purification system (18.2 MΩ·cm, Merck Millipore, Darmstadt, Germany).

#### 2.1.2. NMR Spectroscopy

^1^H-NMR measurements were carried out on a 400 MHz Bruker Avance Neo instrument, Billerica, MA, USA. A mixture of deuterium oxide (D_2_O) and deuterium chloride (DCl 1%) was used as the solvent for the polymers, while deuterated chloroform (CDCl_3_) was employed as the solvent for NCAs, at room temperature.

#### 2.1.3. FT-IR Spectroscopy

Fourier transform infrared (FT-IR) spectroscopy measurements were conducted using a Perkin Elmer Spectrum One instrument (Waltham, MA, USA), in KBr pellets at room temperature, in the 450–4000 cm^−1^ range.

#### 2.1.4. Size Exclusion Chromatography

Size exclusion chromatography (SEC) was employed to determine the *M*_n_ and Ð*M* = *M*_w_/*M*_n_ values. The analysis was performed using two different SEC sets. The first one was composed of a Waters Breeze instrument (Milford, MA, USA) equipped with a 2410 differential refractometer and a Precision PD 2020 two angles (15°, 90°) light scattering detector. The carrier solvent was a 0.10% TFA (*v*/*v*) solution of water/acetonitrile (80/20 *v*/*v*) at a flow rate of 0.8 mL min^−1^ at 35 °C. Three linear Waters hydrogel columns were used as a stationary phase. The second system was composed of a Waters 600 high-performance liquid chromatographic pump, Waters Ultrastyragel columns (HT-2, HT-4, HT-5E and HT-6E), a Waters 410 differential refractometer and a Precision PD 2020 two angles (15°, 90°) light scattering detector at 60 °C. A 0.1 M LiBr in DMF solution was used as an eluent at a rate of 1 mL min^−1^.

#### 2.1.5. Circular Dichroism

Circular Dichroism measurements were conducted via a JASCO J–815 model in a 0.1 cm cell. The aqueous solutions of polymers had a concentration of 10^−5^ g mL^−1^ and the desired pH was adjusted with the addition of droplets either of 0.01 N HCl, or 0.01 N NaOH. The temperature was stabilized to 25 °C with the use of a dedicated digital thermostat. The nitrogen flow was adjusted to 6.0 L min^−1^.

#### 2.1.6. UV Spectroscopy

UV spectroscopy was carried out using a Perkin Elmer Lamda 650 spectrometer, (Waltham, MA, USA) in the range of 250–800 nm, at room temperature, with cells requiring 3 mL. A Waters Diode-Array 690 detector (Milford, MA, USA) was used for the calibration and on-line determination of the DOX drug loading efficiency at λ = 485 nm.

#### 2.1.7. Dynamic Light Scattering

DLS measurements were conducted with a Brookhaven Instruments BI-200SM Research Goniometer system (Holtsville, NY, USA) operating at λ = 640 nm and with 40 mW laser power. Correlation functions were analyzed by the cumulant method and the Contin software. The correlation function was measured at 90°, at 25 °C. All measurements were performed in either an isotonic PBS or Tris buffer (10 mM, 150 mM NaCl) at pH = 7.4, PBS buffer (10 mM, 150 mM NaCl) at pH = 6.5, and an isotonic acetate buffer (10 mM, 150 mM NaCl) at pH = 5.0. The concentration range measured was between 2 × 10^−3^–1 × 10^−5^ g mL^−1^.

#### 2.1.8. Static Light Scattering

SLS measurements were carried out on an ALV/CGS-3 Compact Goniometer System (ALV GmbH, Langen, Germany), equipped with an ALV-5000/EPP multi-tau digital correlator with 288 channels and an ALV/LSE-5003 light scattering electronics unit for stepper motor drive and limit switch control. A JDS Uniphase 22 mW He-Ne laser was used as the light source. The instrument was connected to a Polyscience model 9102 bath for temperature control, allowing measurements at variable temperature.

#### 2.1.9. Electrophoretic Mobility

The electrophoretic mobility measurements of the empty and drug-loaded nanoparticle dispersions were conducted using a Brookhaven Instruments Nanobrook Omni system (Holtsville, NY, USA) operating at λ = 640 nm and with 40 mW power, operating in PALLS mode. All the measurements were performed in isotonic Tris buffer (10 mM, 150 mM NaCl) at pH = 7.4 at 37 °C and were the average of at least three runs.

#### 2.1.10. Transmission Electron Microscopy

Transmission electron microscopy images were obtained using a Jeol 2100 TEM, operated at 200 kV and fitted with a Gatan Ultrascan 1000 camera (Pleasanton, CA, USA). Samples for TEM analysis were prepared via drop-casting a few milliliters of sample dispersions after ultrasonication onto holey carbon grids, allowing the solvent to evaporate and leaving the sample to rest for 24 h at ambient temperature.

#### 2.1.11. Cell Culture

The MCF-7, MDA-MB 231 and T47D cell lines were cultured in RPMI 1640 growth medium, supplemented with 5% fetal bovine serum, 2 mM l-Glutamine, 100 U mL^−1^ penicillin and 100 μg mL^−1^ streptomycin. Cells were maintained at 37 °C in a humidified 5% CO_2_ incubator. ΕR(+) human breast cancer cell line MCF7, ER/PR (+) human breast cancer cell line T47D and triple negative human breast cancer (TNBC) cell line MB231 were purchased from NCI (NCI, NIH, Frederick, MD, USA).

### 2.2. Synthesis of the Monomers

#### 2.2.1. Synthesis of N^im^-Trityl-l-Histidine N-Carboxy Anhydride (N^im^-Trityl-l-His-NCA)

For the synthesis of *N^im^*-*Trityl*-l-His-NCA a previously reported method by our group [16] was mainly followed with some modifications. The synthesis was conducted in two steps. In the first step, the hydrochloric salt of *N^im^*-*Trityl*-l-His-NCA was formed, and finally the desired pure *N^im^*-*Trityl*-l-His-NCA was synthesized after the removal of HCl. The whole synthetic procedure is presented in detail in Appendix A.

#### 2.2.2. Synthesis of S-tert-Butyl-mercapto-l-Cysteine N-Carboxy Anhydride (tBM-l-Cys-NCA)

The *N*-carboxy anhydride of *t*BM-l-Cys-NCA was synthesized in a similar way to a previously described process [37]. The synthetic steps are provided in Appendix A.

#### 2.2.3. Synthesis of Sarcosine N-Carboxy Anhydride (Sar-NCA)

The synthesis of the Sarcosine *N*-Carboxy Anhydride was conducted following previously reported procedures [16,38]. The total synthetic route is described thoroughly in Appendix A.

### 2.3. Synthesis of the Hybrid-Polypeptides

The synthesized polymers are illustrated in Figure 1. The synthetic procedure is described in detail in Appendix A. Briefly, the synthesis of the hybrid polypeptide terpolymers was achieved through a ring-opening polymerization (ROP) process [39,40,41] of the corresponding *N*-carboxy anhydrides, using an amino end-functionalized poly(ethylene oxide) (*m*-PEO-NH_2_) macroinitiator, with molecular weight 10.0 × 10^3^ g mol^−1^. Highly purified DMF was the solvent at all polymerizations. In case of the *m*PEO-*b*-PHis-*b*-PCys as well as the *m*PEO-*b*-PCys-*b*-PHis hybrid terpolymers, the sequential addition synthetic approach of the corresponding anhydrides of the amino acids was used, after the completion of the polymerization of each monomer. In case of *m*PEO-*b*-[PCys-*co*-PHis] terpolymers, the macroinitiator polymerized the mixture of the two anhydrides. Then, the hybrid polypeptides were precipitated followed by deprotection of the trityl group of His by TFA. Finally, the deprotection of cysteine was achieved by using 1,4 dithiothreitol (DTT). A general reaction sequence for the synthesis of hybrid terpolymers of the general type mPEO-*b*-P(Cys)-*b*-P(His) (by the term general type we mean the three different structures with different PCys topology) is given in Figure 2.

Since the PEO blocks were equal for all the polymers, the code of the hybrid polypeptides was defined by the order of the blocks as well as the monomeric units of L-cysteine; therefore, the abbreviation PHis-PCys5 refers to the triblock *m*PEO_227_-*b*-P(His)_40_-*b*-P(Cys)_5_ and PCys10-PHis refers to *m*PEO_227_-*b*-P(Cys)_10_-*b*-P(His)_35_, while in case of the *m*PEO_227_-*b*-[P(Cys)_5_-*co*-P(His)_40_], where the polypeptidic block is composed of randomly distributed peptides, PCys5COPHis will be mentioned.

### 2.4. Self-Assembly of Empty NPs via Solvent Switch Method

The ability of the synthesized polymers to self-assemble in aqueous media was examined at five different isotonic buffers, with different pH values and GSH concentrations. The pH values of 7.4 and 6.5 were adjusted with a PBS buffer solution (10 mM, 150 mM NaCl), while the pH 5.0 was achieved with an acetate buffer solution (10 mM, 150 mM NaCl). At pH 6.5 and 5.0, another two buffers were prepared containing 10 mM of GSH, in order to study the influence of this reducing agent in the self-assembly behavior of the NPs. In a typical procedure, 10 mg of the hybrid polypeptides, as well as 0.02 g of DTT were dissolved in 2 mL of DMSO. After the complete dissolution, 18 mL of MilliQ water were added dropwise, and the whole mixture was left under stirring overnight. The next day, the solution was placed in a dialysis membrane (Spectrapor MWCO 3500 Da) and was dialyzed against 2 L of PBS buffer pH = 7.4 for 3 h. Then, the dialysis membrane was transferred to a fresh media of the same buffer and dialyzed for another 3 h with the presence of 10 mL H_2_O_2_. The last dialysis was lasted 12 h and then the solution of the NPs was collected and divided in 5 equal parts. The first part was kept for DLS measurements, while the remaining solution was transferred equally in four different dialysis membranes (Spectrapor MWCO 3500 Da) and was dialyzed against 2 L of the following buffers for 24 h with frequent changes of the external media: PBS pH = 6.5, PBS pH = 6.5 and 10 mM GSH, pH = 5.0, pH = 5.0 and 10 mM GSH. Finally, the solution of each membrane was collected and measured with DLS, after filtration with a 0.45 μm hydrophilic filter.

### 2.5. Loading of Anticancer DOX

In a typical experiment, 10 mg of the fully deprotected polymers was dissolved in 2 mL of DMSO and left under stirring overnight, to afford clear solutions. In case of the polymers containing PCys, 0.02 g (0.13 mmol, 9:1 mol DTT/mol Cys) DTT was added, in order to avoid the undesirable crosslinking reactions. Subsequently, a special treatment of DOX (HCl-salt) was conducted, according to a standard procedure described by Kataoka et al. [42]. In line with this protocol, 5 mg of DOX hydrochloride was dissolved in 100 mL of MilliQ water, and the resulting red solution was added in a separatory funnel containing 100 mL of chloroform. Then, 3.0 equivalent of triethylamine (TEA) (mol Et_3_N: mol Dox × HCl = 3:1) was added in the aqueous phase and the color immediately turned to purple. After shaking the solution, the color became red again and the DOX was distributed in the organic phase. The concentration of DOX in the aqueous phase was estimated photometrically at 485 nm and the pH measured was close to neutral. Then, the hydrophobic DOX-free base dissolved in chloroform was separated and collected in a flask. The organic solvent was distilled off and the solid DOX was obtained. Afterwards, the solution of each polymer in DMSO was added in the flask containing the dried DOX and was left for half an hour to be dissolved. Then, 8 mL of PBS buffer (pH = 7.4) was added dropwise to the above mixture over a period of 10 min. The solution was then placed in a dialysis bag (Spectrapor, MWCO: 3500 Da, 25 °C) and was dialyzed against 4 L of isotonic PBS buffer at pH = 7.4 (150 mM NaCl, 10 mM PBS), in order to remove the excess drug. After 3 h of dialysis, the external buffer was renewed, and 30 mL of H_2_O_2_ was added in the fresh buffer, in the case of the polymers containing poly(l-cysteine), in order to induce the crosslinking reaction. The dialysis lasted another 3 hours and then the same procedure was repeated, without the addition of H_2_O_2_, for 12 h in total. The next day, the solution inside the membrane was obtained and the volume measured was about 12 mL. Then, about 4 mL of the NP solution was preserved for further analysis and the rest of the solution was divided into five equal parts of 1.5 mL and each part was added in a new dialysis membrane (Spectrapor, MWCO: 6000–8000 Da) and was immediately immersed in 35 mL of buffers with different characteristics, as far as the pH, the temperature and the concentration of GSH are concerned, in order to study the in vitro DOX release profile. The encapsulation efficiency (EE) and the loading capacity (LC) of the different NPs were calculated by UV absorption spectroscopy at 485 nm, as the polymer did not absorb at this wavelength, while free DOX does. Quantification was achieved by calibrating the instrument with dissolved DOX in the corresponding PBS buffer.

The encapsulation efficiency and the loading capacity were calculated according to the following equations:EE (%) = (mass of Dox in NPs/ mass of Dox in the initial solution) × 100
LC (%) = (mass of Dox in NPs/ polymer mass) × 100

### 2.6. In Vitro Drug Release Studies

In vitro DOX release experiments were conducted at three different pH values (pH = 7.4, 6.5 and 5.0), at two temperatures (37 °C and 40 °C) and as far as the polymers with poly(l-cysteine) in their polypeptidic block are concerned, the factor of the addition of GSH was studied. More precisely, after the completion of the dialysis procedure, the remaining solution of NPs was divided into five equal parts of 1.5 mL, as mentioned above, was transferred into a new dialysis bag (Spectrapor, MWCO: 6000–8000 Da) and was immediately immersed in 35 mL of each of the in vitro release medium. The first membrane was ingrained in a PBS buffer at pH = 7.4 and at 37 °C (0.010 M PBS, 0.150 M NaCl) under stirring at 200 rpm. For the release studies at the acidic pH (6.5 and 5.0), two different samples were employed, for each pH value. The first dialysis bag was immersed in a PBS buffer at pH = 6.5, at 40 °C, (0.010 M PBS, 0.150 M NaCl) and the other was introduced into the same release medium containing 10 mM of GSH. Similarly, in the case of the pH = 5.0, the first membrane was added in an acetate buffer at pH = 5.0, at 40 °C, (0.010 M acetate, 0.150 M NaCl), and the last was sank into the same buffer, at the same conditions with further addition of 10 mM GSH. The cumulative release of the drug was measured at the exterior solution at defined time intervals. The dialysis membrane was transferred into a fresh buffer solution at every interval, in order to avoid saturation of the solution from the hydrophobic drug. The DOX concentration was calculated by UV spectroscopy at λ = 485 nm, using a calibration curve obtained with solutions of known DOX concentration measured using the same instrument.

### 2.7. In Vitro Cytotoxic Activity: Sulforhodamine B (SRB) Assay

The established human cell lines from breast cancer MCF-7 (estrogen and progesterone receptor positive invasive ductal carcinoma), T-47D (progesterone receptor positive invasive ductal carcinoma) and MDA-MB231 (triple negative breast cancer) were used and provided by the pharmacology laboratory of NCI (National Cancer Institute, NIH, Frederick, MD, USA).

Cell culture was performed in RPMI 1640 medium (Gibco^®^, Code: 31870025) supplemented with 5% fetal bovine serum (FBS: fetal bovine serum, (Biosera, Code: 1001G)), 2 mM L-glutamine (Biosera, Code: XO-T1715), 100 U mL^−1^ penicillin and 100 μg mL^−1^ streptomycin (Biosera, Code: XO-A4122). The cell cultures were kept in an incubation oven, at 37 °C, in an atmosphere of 5% CO_2_ and 95% humidity.

The antiproliferative activity of the NPs was tested by the colorimetric method of SRB [43,44]. SRB is an anionic micromolecular compound that is stoichiometrically attached to the basic amino acid residues of protein chains, under slightly acidic conditions, and then extracted, under basic conditions.

This process involves the following steps. At the beginning of each experiment, the viability of the cells is checked with the trypan blue method so that it is always greater than 96%. The cells are added to 96-well flat-bottom cell culture plates (density 5000–10,000 cells per position) and incubated for 24 hours in an incubation oven at 37 °C, 5% CO_2_ and 95% humidity to return to the logarithmic development phase (adjustment period). After 24 hours, the NP solutions are added. In some cells, only culture material is added to provide the control cells (control, C). Each NP solution was tested in four logarithmic concentrations with a maximum concentration of 10 μM. The final concentration of DMSO in each cell culture was not higher than 0.1%. A number of sites from each cell line in each experiment are fixed with 50% *v*/*v* TCA (Trichloroacetic Acid) (Applichem, Code: A1431) cold solution for 1 h at 4 °C, after 24 hours of the adjustment period, aiming the representation of cell culture in the phase of addition at NPs (Tz). After 48 hours of incubating the cells with the NPs, the cells are fixed by gently adding 50% *v*/*v* TCA to each site of the cell culture plate, for 1 hour, at 4 °C. The cells are then carefully washed, 3 times, with deionized water, the excess water is removed and the plates are allowed to dry at room temperature. The cells are stained with a solution of 0.04% *w*/*v* SRB (from SIGMA, Code: S9012) in 1% acetic acid (from Fluka, Code: 45731), for 10 minutes, at room temperature. After incubation, the excess dye is removed by repeated rinsing with 1% *v*/*v* acetic acid and the cell monolayers are allowed to dry at room temperature. A 10 mM Tris base solution is then added and the cells are incubated for 10 minutes at 37 °C. Under these conditions, the protein-bound dye is released into the slightly basic Tris base solution. For each concentration of the studied NP solution, the optical absorption at 540 nm (Ti) is measured on a BioTek microplate reader (Biotek, EI-311).

Using the optical absorption measurements of the cells at the time of addition of the NPs (Tz), the control cells (C), as well as the cells under the influence of the examined NPs, the percentage growth of the cells (% growth rate) can be calculated with the use of the following equations:[(Ti − Tz)/(C − Tz)] × 100, for concentrations where Ti ≥ Tz and 
[(Ti − Tz)/Tz] × 100, for concentrations where Ti < Tz

From the resulting dose–response curves (response, the cell growth rate, % growth rate) the parameters GI50, TGI and LC50 are determined, where:

GI50, Growth Inhibition 50% = the concentration of the drug through which cell growth is inhibited by 50%.

TGI, Total Growth Inhibition = the concentration of the drug through which total inhibition of cell growth is achieved.

LC50, Lethal Concentration 50% = the concentration of the drug that causes death in 50% of the cell population [45,46].

## 3. Results and Discussion

### 3.1. Synthesis and Characterization of the N-Carboxy Anhydrides (NCAs)

The synthesis of the *N*-carboxy anhydrides of *α*-amino acids was monitored by FT-IR spectroscopy, while the successful synthesis and the high purity of the final monomers were confirmed by ^1^H-NMR spectroscopy. The results from the characterization of the *N*-carboxy anhydrides are summarized in Appendix A (Appendix A, Appendix A).

### 3.2. Synthesis and Characterization of the Polymers

Initially, the novel fully protected polymers of the general type of PEO-*b*-P(*N^im^*-*Trityl*-l-His)-*b*-P(*t*BM-l-Cys) were synthesized followed by the selective deprotection of each polypeptide block, to afford the fully deprotected polymers of the general type of *m*PEO-*b*-P(Cys)-*b*-P(His). The synthetic procedure was monitored by FT-IR spectroscopy, and the molecular weights were obtained by using SEC-TALLS while the controlled cleavage of the protective groups was confirmed by ^1^H-NMR. The polymers were excessively characterized and the characterization results are shown in Table 1. It can be seen that the novel hybrid terpolymers exhibited a high degree of molecular and compositional homogeneity, while the experimentally obtained molecular characteristics were within 10% close to the stoichiometric one. In addition, the total molecular weight of the polypeptidic blocks were close in all polymers although the ratios between the PHis and PCys were different, while PEO was the same.

As an example, we will present the characterization of the PCys5-PHis. The FT-IR spectra of the block copolymer *m*PEO_227_-*b*-P(*t*BM-l-Cys)_5_-*b*-P(*N^im^*-*Trityl*-l-His)_40_ is presented in Appendix A. Spectrum A corresponds to the protected copolymer *m*PEO_227_-*b*-P(*t*BM-l-Cys)_5_. The vibration at 1637 cm^–1^ is attributed to the C=O bond of the amide bond. Other characteristic peaks appear at 1100 cm^–1^ and 2890 cm^–1^, which are due to the amplitude vibration of the ether bond C–O–C of PEO and C–H bonds respectively. In addition, the vibration at 1740 cm^–1^ corresponds to the vibration of one carbonyl group of l-cysteine *N*-carboxy anhydride, which indicates that the polymerization of the first monomer has not been completed at the time of the measurement. Spectrum B (Appendix A) corresponds to the copolymer *m*PEO_227_-*b*-P(*t*BM-l-Cys)_5_-*b*-P(*N^im^*-*Trityl*-l-His)_40_ and was obtained approximately 14 days after the addition of the second monomer (*N^im^*-*Trityl*-l-His NCA). In this spectrum, the characteristic peak at 1679 cm^–1^ is observed, which corresponds to the amide bond, as well as the absorption bands at 1106 cm^–1^ of PEO. Additional peaks that appear in this spectrum are the vibration at 1784 cm^–1^, which is due to the one carbonyl group of *N^im^*-*Trityl*-l-His NCA, and indicates that the polymerization of the PHis block has not yet been completed. Additionally, the peaks at 745 cm^–1^ and 703 cm^–1^ are attributed to the bending vibrations of the -CH=CH- bonds of the aromatic rings of the *trityl* protecting groups of PHis. Spectrum C corresponds to the final fully protected block copolymer, which was isolated after precipitation in diethyl ether. This spectrum shows exactly the same absorption bands as spectrum B, with the difference that the vibration at 1784 cm^–1^ of the anhydride is absent, as the histidine monomer is completely consumed. Finally, the spectrum D corresponds to the desired fully deprotected PCys5PHis. In this spectrum the vibrations at 746 cm^–1^ and 702 cm^–1^, which correspond to the *trityl* protecting groups of the poly(l-histidine), are absent, proving the successful deprotection of this polypeptide block. However, there are no accurate data from the FT-IR spectrum to verify the successful deprotection of poly(l-cysteine) building blocks, as the vibration signals of -SS-, -CH_2_-S-, -SH, -CH (*t*-butyl) bonds are very weak. The successful synthesis and the purity of the terpolymer was confirmed by ^1^H-NMR spectroscopy in D_2_O/DCl 1% solvent, after each deprotection step. It is observed that all peaks in both spectra (Appendix A) are attributed to the hydrogens of the polymer. In Appendix A, the upper spectrum corresponds to the histidine-deprotected *m*PEO_227_-*b*-P(*t*BM-l-Cys)_5_-*b*-P(His)_40_, while the second is attributed to the final fully deprotected PCys5PHis. Appendix A, Spectrum A: ^1^H-NMR (600 MHz, D_2_O/DCl 1%, δ, ppm): 1.34 (i: 9H, (CH_3_)_3_–C–), 3.18–3.24 (f + g: 4H, –CH_2_–), 3.41 (h: 3H, CH_3_–O–), 3.35–3.90 (e: 4H, –CH_2_–CH_2_–O–), 4.40 (d: 1H, NH–CH(CH_2_–S–S)–C=O), 4.77 (c: 1H, NH–CH(CH_2_–Im)–C=O), 7.35 (b: 1H, –C=CH–N–), 8.71 (a: 1H, –N=CH–N–). Appendix A, Spectrum Β: ^1^H-NMR (600 MHz, D_2_O/DCl 1%, δ, ppm): 1.35 (i: 9H, (CH_3_)_3_–C–), 3.20–3.24 (f + g: 4H, –CH_2_–), 3.42 (h: 3H, CH_3_–O–), 3.35–3.90 (e: 4H, –CH_2_–CH_2_–O–), 4.53 (d: 1H, NH–CH(CH_2_–S–S)–C=O), 4.75 (c: 1H, NH–CH(CH_2_–Im)–C=O), 7.36 (b: 1H, –C=CH–N–), 8.71 (a: 1H, –N=CH–N). Finally, the histidine-deprotected terpolymer *m*PEO_227_-*b*-P(*t*BM-l-Cys)_5_-*b*-P(His)_40_ was characterized by SEC chromatography in H_2_O/TFA solvent (Appendix A).

A similar procedure was followed for the synthesis and characterization of all hybrid terpolymers. The characterization results from all polymers obtained by FT-IR, ^1^H-NMR and SEC are presented in Appendix A (Appendix A).

### 3.3. Secondary Structure through Cyclic Dichroism

It is well known that polypeptides have the ability to mimic natural proteins by adopting secondary structures in response to various external stimuli (temperature, pH, etc.). 

In order to investigate their structural and conformational changes by pH and temperature, we studied the synthesized polymers by CD. More precisely, CD measurements were conducted at four different pH values: pH = 7.4 (pH of the healthy tissue), pH = 6.5 (pH of the extracellular environment of the tumor tissue as well as early endosome pH within the cells), pH = 5.0 (pH of the lysosomes within the cell) and pH =3.0, and at three different temperatures: 25 °C (room temperature), 37 °C (temperature of the healthy tissue) and 40 °C (temperature of cancer tissue). We studied the PCys-protected polymers, while only PHis was deprotected, in order to avoid crosslinking.

In most cases, the results revealed a similar conformational transition of the secondary structure from a beta turn at higher pH values (pH = 7.4 and pH = 6.5) to a random coil conformation at lower pH values (pH = 5.0 and pH = 3.0), as shown in Figure 1 as well as Appendix A. The negative peaks at 190 in combination with the positive peak at 205 nm and a slight negative peak at 218 nm are indicative of the *β*-turn type 2 conformation [47], the negative peak at 218 and a positive at 195 nm reveal a *β*-sheet conformation, while the negative peak at 225 nm is indicative of an *α*-helix conformation. At lower pH, the negative peak at 196 nm in combination with the positive peak at 218 nm are characteristic of the random coil structure.

As we showed in our previous work, at higher pH, the *β*-turn is a conformation that enthalpically favors the structure of PHis homopolymer [48]. In this conformation, the imidazole rings come close, developing the maximum hydrogen bonds. In this structure, a loop is created every three amino acids, since the nitrogen of the imidazole ring of an amino acid forms hydrogen bonds with the carbonyl group of the following amino acid and at the same time, forms hydrogen bonds with the hydrogen of the imidazole ring of the following amino acid. At lower pH, the secondary structure changes from *β*-turn to random coil conformation.

The conformational transitions obtained by altering the pH (*m*PEO_227_-*b*-P(*t*BM-l-Cys)_10_-*b*-P(His)_35_ (Figure 1c), *m*PEO_227_-*b*-P(His)_40_-*b*-P(*t*BM-l-Cys)_5_, *m*PEO_227_-*b*-P(His)_35_-*b*-P(*t*BM-l-Cys)_10_ and *m*PEO_227_-*b*-[P(*t*BM-l-Cys)_5_-*co*-P(His)_40_]) (Appendix A–S34, see supporting information) are similar to the one obtained by PEO-*b*-PHis diblock copolymer (Figure 1d). At the triblocks, a lower pH is required as compared to the one required for PEO-*b*-PHis to achieve the transition to the random coil conformation, due to the higher amount of hydrophobic blocks. At these terpolymers, the absorption of the PHis block dominates and overlaps the absorbance of the *β*-sheet conformation of the protected PCys block. However, in some cases, the secondary structure of the *β*-sheet is evident, which is the typical conformation of the free and protected PCys [49]. The terpolymer PEO-*b*-P(*t*BM-l-Cys)_5_-*b*-P(His)_40_ (Figure 1a) at pH = 7.4 exhibits a mixed structure of *β*-turn and *β*-sheet, which is attributed to both the PHis and PCys moieties. At pH = 6.5, a mixed structure of *β*-turn and *α*-helix is observed, as we can see a negative peak at 190 and a positive at 205 nm, while we also observe a negative peak at 230 nm indicative of the *α*-helix conformation. Finally, at more acidic pH (pH = 5.0 and pH = 3.0), only the conformation of the random coil is observed. 

In case of *m*PEO-*b*-(P(His)_35_-*co*-P(*tBM*-L-Cys)_10_) (Figure 1b), the presence of a larger amount of PCys randomly distributed along the PHis chain induces a larger amount of *β*-sheet conformation at neutral pH. This is more pronounced at this copolypeptide due to the higher amount of PCys (Figure 1b) rather the one with lower amount and the same structure (random distribution of PCys) (Appendix A). It seems that the small amount of PCys did not have significant impact on the secondary structure. The difference of the secondary structure obtained from the random as compared to the block copolypeptides is proof of the random distribution of PCys along the PHis chain on the PCys5COPHis and PCys10COPHis terpolymers. Finally, it was found that by increasing the temperature maintaining a constant pH, from 25 °C, to 37 °C and then to 40 °C, the conformation is not altered (Appendix A), which shows that the polymers do not show a temperature responsiveness. 

### 3.4. Self-Assembly of the Empty Hybrid Polymers

The ability of the synthesized polymers of the general type *m*PEO-*b*-P(Cys)-*b*-P(His) to form nanostructures was achieved via a solvent switch method, by applying the dialysis technique, with the use of DMSO as the common good solvent and aqueous solution at pH = 7.4, as the final media. During this procedure, a simultaneous crosslinking reaction was conducted, using H_2_O_2_ as the oxidative agent to form the disulfide bonds which stabilize the NPs. The ability of the polymers to self-assemble as well as the structural characteristics of the NPs were examined by DLS, SLS and TEM. At pH = 7.4, in which the self-assembly and crosslinking takes place, two populations are always observed by DLS (Appendix A). The average size of the small population is about 30 nm, while the larger population is about 250 nm. The appearance of two populations is also observed by TEM microscopy. More specifically, for the crosslinked polymer PCys5-PHis the TEM image depicted at Figure 2g shows small spherical and elliptical vesicles within the core of a larger spherical nanostructure. The large NPs are composed of a large core containing multiple small vesicles. The matrix of the core is a mixture of PHis and PCys. Therefore, the dimensions of the NPs obtained by DLS (~31 nm, Appendix A) are the small vesicles within the core of the large NPs shown as the second population. The core of the NPs obtained by TEM for these NPs is almost 210 nm but if we add the PEO corona, we will achieve dimensions close to 220 nm which are smaller than the 250 nm obtained by DLS (Appendix A). This is probably due to the different processes followed for the sample preparation for DLS and TEM. For DLS, the nanoparticles are dissolved in PBS buffer pH = 7.4, while for TEM imaging, the treatment includes the removal of the salts by dialysis, freeze drying and finally redissolution in MilliQ water to be placed on the grid and the final evaporation of water to dryness. This difference in dimensions obtained between the two methods is common in many works [28], where TEM gives smaller dimensions, and can be attributed to the shrinkage caused by the evaporation of water. No TEM measurement was performed for the crosslinked polymer PCys10-PHis, as it precipitated during the process of self-assembly and crosslinking at PBS buffer pH = 7.4. The precipitation is due to the increased hydrophobicity and crosslinking, which is a consequence of the higher percentage of PCys in the polypeptide block and the close packing of PCys since they are obliged to be organized and located at the interphase between the PEO and PHis phases. In all cases, TEM measurements (Figure 2) revealed spherical micellar structures, with a multivesicular core comprised of PHis and PCys polypeptides and a hydrophilic corona of PEO. This kind of self-organization is consistent with the results from DLS and SLS, as mentioned.

SLS measurements confirm the self-assembly of the NPs in structures containing a multivesicular core, as the ratio R_g_/R_h_ is close or slightly larger than 1. Table 2 summarizes the polymers of the present work and the corresponding values of the sizes R_g_, R_h_ and R_g_/R_h_, at pH = 7.4, at 25 °C.

Finally, a general observation concerning all NPs is that in TEM images, around the gray core, a faint, white crown can be seen, which is attributed to the PEO block, as it does not create a strong contrast. This phenomenon comes in agreement with the results from z-potential measurements (Table 2), which reveal that at pH = 7.4, the mean value of the z-potential is in the range [−6.8 mV, + 3.3 mV], therefore, all the synthesized NPs have in most cases a neutral surface charge, indicating that the PEO block consists of the outer periphery of the nanostructures.

### 3.5. pH and Redox Responsiveness of the Empty NPs

In order to investigate the pH and redox responsiveness of the synthesized NPs at both healthy and cancerous tissue conditions, different aqueous solutions of the polymers were prepared and measurements took place at different pH values and GSH concentrations. More specifically, DLS measurements were conducted to the solutions of the empty crosslinked polymers, resulting from the dialysis process, at pH = 7.4 (pH of human blood and healthy tissues), at pH = 6.5 (extracellular pH of cancer cells and early endosomes inside the cells) and at pH = 5.0 (lysosomal pH inside the cells), as well as at pH = 6.5 and pH = 5.0 with the addition of 10 mM GSH (intracellular GSH of cancer cells). 

DLS measurements revealed responsiveness towards pH and GSH concentration. In the case of PCys5-PHis NPs (Appendix A), at pH = 7.4, the results indicate the existence of two populations, one small of about 31 nm and a larger one of about 250 nm. With a decrease in pH from 7.4 to 6.5 and 5.0 (25 °C, 90°), a slight increase in the diameter is observed, due to swelling of PHis through interaction with water, since at this pH PHis is protonated, which in turn leads to an increase in its hydrophilicity.

In the presence of 10 mM GSH at the acidic pH, the NPs exhibit a further redox response. GSH acts as a reducing agent and causes the cleavage of the disulfide bonds. The concentration of GSH is about 10–20 mM in cancer cells, while it is about 2 μΜ in healthy tissues. It is observed that at pH = 6.5, in the presence of GSH, a third population appears at 7 nm, most likely due to the rupture of the NPs to smaller particles or even single chains. In addition, at pH = 5.0 in the presence of GSH, except for the third population that appears at 7 nm, there is an additional increase in the size of the larger population from 285 nm (pH = 5.0, without GSH) to 427 nm (pH = 5.0, with GSH). Both of these results prove the synergistic response of PHis and PCys, through the variation in pH and GSH under healthy and cancerous conditions. The same trend is observed for all the polymers and the results from DLS measurements are summarized in Appendix A (Appendix A).

### 3.6. Self-Assembly of the DOX-Loaded NPs

The encapsulation of DOX was performed on all series of the polymers, as well as on another two polymers PEO_227_-*b*-P(His)_44_ and PEO_227_-*b*-P(Sar)_98_-*b*-P(Cys)_30_. DLS and TEM techniques were employed to obtain the structure and the morphological characteristics of NPs, while z-potential measurements were conducted to determine their surface charge. All the results are summarized in Table 3 and Figure 3, while DLS results are presented in Appendix A (Appendix A).

Surprisingly, the results from DLS and TEM revealed that in some cases, the DOX-loaded NPs can self-assemble into homogeneous core–shell micellar structures, instead of polydisperse micellar structures with a multivesicular core obtained by the empty one. It can be seen (Table 3) that the diameter of the NPs obtained by TEM is smaller than the R_h_, due to the different preparation methods followed, as referred to previously. As an example, the TEM images and the size distribution of the core of the NPs obtained by the PCysX-PHis, are shown in Figure 3b–d. It is obvious that the NPs formed by these terpolymers are core–shell micelles, composed of a core containing the hydrophobic polypeptides along with encapsulated DOX, with a corona composed of a PEO chain and water. The dimension of the NPs obtained by DLS is close to 190 nm, while the one obtained by TEM is close to 110 nm. In case of the TEM images, we see only the core of the NPs, while by DLS we see the outer dimensions of the NPs. Still, the addition of the PEO which is smaller than 10 nm on the dimensions obtained by TEM cannot match the dimensions of the DLS. The structure of the micelles is illustrated in Figure 3n. 

The TEM images of the DOX-loaded NPs of the PHis-PCysX as well as the size distribution of the core of the NPs are shown in Figure 3f–i. The NPs are vesicular structures of rather small dimensions. In these images, a faint white diffuse cloud is observed that surrounds the vesicle and is due to the hydrophilic block of PEO. The possible structure of the vesicles is illustrated in Figure 3o. Particularly at these two polymers, the significant difference between the DLS and TEM dimensions is probably due to the formation of vesicular structures, where the elimination of the solvent is expected to result in a significant reduction in dimensions (as observed by TEM) due to the shrinkage of the NPs. 

The TEM images of the DOX-loaded NPs of PCysXCOPHis are depicted on Figure 3a,e. The TEM image of PCys5COPHis in Figure 3ashows ruptured aggregates, probably ruptured vesicular structures. It seems that the small amounts of PCys at this polymer did not result in efficient crosslinking that would stabilize the aggregate. It seems that the random distribution of the small amount of PCys did not result in efficient crosslinking. On the contrary, in the case of the PCys10COPHis with higher amount of Pcys (Figure 3e), the NPs were more robust and aggregates with a core containing smaller vesicles were formed. The vesicular structures within the core were very small and could not be distinguished by DLS. In that series of NPs, the dimensions obtained by DLS and TEM are close.

The TEM image and the core size distribution of the DOX-loaded NPs of *m*PEO_227_-*b*-P(His)_44_ are shown in Figure 3j,k, respectively, while the TEM of the *m*PEO_227_-*b*-P(Sar)_98_-*b*-P(Cys)_30_ is shown in Figure 3l. Both NPs self-assemble into spherical core–shell micellar structures. The hydrophobic core of the micelle, where DOX is encapsulated, consists of the blocks of PHis and PCys for the polymers *m*PEO_227_-*b*-P(His)_44_ and *m*PEO_227_-*b*-P(Sar)_98_-*b*-P(Cys)_30_, respectively. The outer hydrophilic corona of the micelles is attributed to the PEO block in the case of *m*PEO_227_-*b*-P(His)_44_ and to the PEO and poly(sarcosine) blocks for the polymer *m*PEO_227_-*b*-P(Sar)_98_-*b*-P(Cys)_30_. It is obvious that the presence of PCys at the terpolymers significantly altered the structure of the NPs as compared to the one that lacked the PCys layer. 

Finally, Z-potential measurements were conducted in all DOX-loaded NPs, in order to determine their surface charge. The results are presented in Table 3 and show that at pH = 7.4, the mean value of the z-potential is in the range [−8.1 mV, + 1.1 mV], concluding that all synthesized NPs have a neutral surface charge. These results come in accordance with the observations from TEM images, which prove that the uncharged and hydrophilic block of PEO is located at the outer periphery of the nanoparticle. In summary, TEM imaging revealed how the PCys topology as well as the encapsulation of DOX affects the morphology of the DOX-loaded NPs. Thus, it is expected that the topology of the polypeptidic blocks will influence the kinetics of drug release under healthy and cancer cell conditions.

### 3.7. Drug Loading and In Vitro Release Studies

Drug loading was performed at pH = 7.4, using PBS isotonic buffer (150 mM NaCl, 10 mM PBS). The encapsulation efficiency (EE) of the drug and the loading capacity (LC) of the various NPs were determined by UV–Vis spectrophotometry at 485 nm, since only DOX absorbs in this wavelength. Quantification was performed using a standard DOX calibration curve in the corresponding PBS buffer pH = 7.4, presented in Appendix A (Appendix A). Table 3 summarizes the results from UV–Vis spectroscopy measurements.

The drug release profile was examined at various pHs, temperatures, and in the presence of GSH, in order to simulate the release conditions in both healthy (pH = 7.4, 37 °C) and cancer tissue (pH = 6.5, 40 °C, 10 mM GSH) as well as late lysosomes environment of the cancer cells (pH = 5.0, 40 °C, 10 mM GSH). The amount of DOX released was determined by UV–Vis spectrophotometry at 485 nm and quantification was performed using standard calibration curves of the drug in the respective buffers, presented in Appendix A (Appendix A).

It can be seen that the NPs consisting of the aggregated polymer PCys5-PHis in Figure 4a are pH-stimuli responsive, since after 144 hours, 34% of the drug has been released at pH 7.4, 56% at pH 6.5 and 60% at pH 5.0. It is obvious that as the pH of the release medium decreases, the percentage of released DOX increases. This effect is expected, since at acidic pH, the imidazole ring of histidine is protonated, rendering the PHis blocks hydrophobic, leading to the swelling or the rupture of NPs. This is in agreement with the DLS results. Finally, the rupture of the nanoparticles leads to the release of DOX in a pH-controlled manner. It seems that PCys is not contributing significantly to the release of the drug, since the lowering of only the pH results in a significant increase at the release rate of DOX. Therefore, 5 monomeric units of Cys is not enough to create a strong crosslinked layer that will direct the release of the drug.

Contrary to the NPs formed by PCys5-PHis with the lower amount of PCys, the NPs formed by the polymers exhibited the same architecture but higher PCys amount, i.e., PCys10-PHis, at 144 hours, only 15% of the drug was released at pH = 7.4, 29% at pH = 6.5, while at pH = 5.0, 79% was released (Figure 4b). 

It is worth noting that the percentages of the drug released at pH = 7.4 and 6.5 from the PCys10-PHis NPs are lower obtained at all the NPs. This may be due to the presence of the crosslinked PCys layer at the interphase of PEO that maintain the cargo within the core until it is heavily ruptured by an increased concentration of GSH (see Figure 3n). At the same time, the greater stability of the NPs due to the more extensive crosslinking leads to a more pronounced response to the GSH concentration, as at pH = 6.5 without GSH, the release reaches 29%, and at the same pH in the presence of GSH, the percentage increases to 45%. At pH = 5.0, without GSH the release reaches 75%, while at pH = 5.0 in the presence of GSH, it reaches 84%, demonstrating a strong synergistic response to these stimuli (pH and GSH). It seems that the strong swelling of the core at pH = 5.0 even without GSH leads almost to the rupture of the PCys crosslinked layer.

For the DOX-loaded NPs of PHis-PCys5 hybrid terpolymer, pH plays a more critical role than GSH. The NPs formed are micellar structures with a multivesicular core that are expected to be less robust than the core–shell micelles. As in the case of the PCys5-PHis, by lowering the pH we have a significant release of the drug (Appendix A). Due to the formation of a PCys monolayer within the bilayer of the hybrid copolypeptide (see Figure 3o), the release of the drug at pH = 7.4 remains rather low (about 25%) but increases significantly by the protonation of PHis when the pH is lowered. This can be attributed to the transition of PHis from hydrophobic to hydrophilic, and since most of the drug exists within the PHis layer, this switch leads to increased release.

In case of PHis-PCys10 NPs, it was found that the vesicles formed release a significant amount of drug at pH = 7.4. The release become gradual by lowering the pH as well as the increased concentration of GSH. In this case both parameters contribute equally. The maximum cumulative release was 65% (Figure 4c).

For the DOX-loaded PCys5COPHis NPs, core–shell structures composed of a multivesicular core were probably formed that were not robust as indicated by the rupturing under vacuum to dryness (Figure 3a). We have a significant release of drug even at pH = 7.4 which increases gradually by the lowering of pH (Appendix A) and the addition of GSH. The maximum cumulative release obtained was 75%. This release profile is similar to the one of PHis-PCys10, where both stimuli, pH and redox, contribute equally.

Slower release was obtained by the NPs formed by PCys10COPHis hybrid copolypeptides (random structure) (Appendix A), due to the presence of a larger amount of PCys that hinders the release of the drug. The NPs have a similar core–shell structures exhibiting a multivesicular core similar to the one with the lower amount of PCys and the same structure. Although the release at pH = 7.4 is rather high, there is a gradual increase in the release by lowering the pH and addition of GSH. Similar gradual release profile was obtained by the vesicular structures of PHis-PCys10 as well as PCys5COPHis NPs. 

In order to elucidate the influence of the PCys on the release profile of the loaded NPs, we studied the encapsulation and release of DOX of the NPs obtained by the hybrid copolypeptides *m*PEO_227_-*b*-P(His)_44_ and *m*PEO_227_-*b*-P(Sar)_98_-*b*-P(Cys)_30_ shown in Figure 3d,e, respectively. The DOX loaded NPs of the polymer *m*PEO_227_-*b*-P(His)_44_ are core–shell micelles. They show responsiveness only to pH due to the PHis block. Thus, at 144 hours 35% of the drug has been released at pH = 7.4, 63% at pH = 6.5 and 75% at pH = 5.0. It is obvious that although the cumulative release at pH = 7.4 is comparable with most of the NPs in this work, at pH = 6.5 and 5.0 the release is higher. The lower release rates of the DOX at the terpolymers is due to the contribution of the hydrophobic PCys and its crosslinking. At pH = 7.4 where both polypeptides are hydrophobic, the release profile do not depend on the presence of PCys significantly, unless PCys is located at the interphase of PEO. At lower pH, the NPs from *m*PEO_227_-*b*-P(His)_44_ lose their structure faster than the one containing PCys due to the crosslinks formed by this amino acid that stabilize the structure and a lower pH is required to reach the same release rate.

In order to compare the release profile of the terpolymers with the one containing only a hydrophilic polymer and PCys, we synthesized many block polypeptides, initially PEO-*b*-PCys_44_. However, the polymer was not soluble, and we found that the best solubility was on the triblock terpolymer *m*PEO_227_-*b*-P(Sar)_98_-*b*-P(Cys)_30_. The increase in PEO did not result in significant enhancement of the solubility of the NPs, and we incorporated PSar for that purpose. It was found that the NPs formed by the *m*PEO_227_-*b*-P(Sar)_98_-*b*-P(Cys)_30_ hybrid copolypeptide showed a significant release even at pH = 7.4 and inability to maintain the cargo even at neutral pH. In addition, the response to GSH was very strong, while to pH, it was minimal. The weak dependence of the release to pH is due to the protonation of DOX at lower pH which renders it more hydrophilic and not to the polymer. 

In order to further examine the influence of the complex media of RPMI + FBS that mimic an even closer environment at the blood compartment, we performed drug release profiles in this media. To our knowledge, this is the first time that release curves have been performed at the cell culture medium and not only in buffers (Figure 4f). It is obvious that the release profiles are similar to that of the isotonic PBS buffer at pH = 7.4. The results show that even after 2 days, most of the drug is still encapsulated into the NPs and thus, the delivery of the drug is directed by the carriers.

### 3.8. In Vitro Cytotoxic Activity

The antiproliferative activity of the various NP solutions was tested by the colorimetric method of sulforhodamine B (SRB, Sulfurhodamine B). 

From the three cell lines tested, the most sensitive to both DOX and nanoformulations were found to be MCF-7 cells, followed by T-47Ds, while MDA-MB231 were found to be the least sensitive under the experimental conditions used (Figure 5 and Table 4). Interestingly, DOXIL (or CAELYX) does not work well in these experimental conditions, which is probably explained by its composition. In contrast to DOXIL, the four nanocarriers tested showed similar activity to DOX, as shown by both the three cell line growth curves (Figure 5) and the GI50, TGI and LC50 parameters (Table 4). The *m*PEO_227_-*b*-P(His)_44_ NPs showed a slightly better effect on MB231 cells at a concentration of 1 μM (Figure 5); however, the other four nanocarriers did not exhibit any specificity in the cytotoxic activity, similarly to free DOX (Table 4). Finally, none of the empty nanocarriers tested in the same cell lines and experimental conditions showed toxicity.

### 3.9. Influence of the PCys Topology on Self-Assembly, DOX Loading, In Vitro Release Profile as Well as In Vitro Cytotoxic Activity

From the systematic study of the series of hybrid polymers, it is obvious that the topology of PCys plays a critical role in the structure of the NPs formed and thus, the release profile of the drug.

Concerning the empty NPs, the self-assembly resulted in the formation of micellar NPs composed of a multivesicular core and a shell from PEO chains. 

The loaded NPs with DOX presented differently structured NPs as compared to the empty one, which differs depending on the topology of PCys. In the case that PCys is at the interphase between the PEO and PHis, the formed loaded NPs are core–shell micelles. The crosslinked PCys interphase tightly close the drug within the core preventing its leakage at healthy tissue conditions. The drug is released slowly at extracellular cancer pH conditions, while it is released fast and efficiently under intracellular cancer cell conditions, where we have a combination of low pH and high concentration of GSH. Under intracellular healthy conditions, the release is slower than intracellular cancer cell conditions, but slightly higher than under extracellular healthy conditions.

When PCys is at the edge of the polymeric chain, PCys which is more hydrophobic than PHis at neutral pH, interacts with DOX directing the aggregation. So, the chains aggregate first through the PCys end block creating a bilayer with two PEG hydrophilic layers at the outer part leading to the formation of vesicular structures. As illustrated in Figure 3o, the PCys (magenta) layer aggregates in an antiparallel manner to form the bilayer, leading to the formation of vesicular structures. However, due to the encapsulation of DOX also at the PHis layer, we have a significant initial release even at higher pH values, since the crosslinked PCys layer does not hinder the leakage of the drug as in the case when PCys was at the interphase between PEO and PHis (Figure 3n). Still, the release is gradual and responds at both stimuli (pH and redox) when the amount of PCys is higher.

When the PCys is randomly distributed along the PHis chain, the formed loaded NPs are core–shell structures with a multivesicular core. They present a significant initial release at higher pH as in the case of the NPs where the PCys was at the edge of the polymeric chain and the amount of PCys was large. In that case, PCys do not form a crosslinked tight layer, since the monomeric units of Cys are not close together. 

This work shows that it is possible to select a drug release profile and the structure of the NPs formed by altering the topology of PCys. In most cases, both stimuli were participating at the release profile of the NPs. Slow release can be achieved by placing PCys at the interphase of the NPs and will be released fast when the NPs reach an intracellular cancer cell environment. When PCys is located at the edge of the polymeric chain, the NPs will form vesicular structures and it will be possible to encapsulate both a hydrophobic drug within the bilayer and a hydrophilic drug at the empty interior. The release will be performed in a gradual way, by lowering the pH and increasing GSH concentration. In that case a significant amount of PCys has to be incorporated at the NPs. 

A similar gradual release can be achieved when PCys is randomly distributed along the PHis chain. This release profile can be achieved even for low amounts of PCys and through a core–shell micelle structure exhibiting a multivesicular core. 

Usually, the nanoparticulate drug delivery results in a higher cancer cell growth rate as compared to the corresponding growth rate of the free drug [50]. In our work, the cell culture results showed that the GI50 of the NPs is comparable or better to the free drug after two days, although at least half of the drug is still encapsulated within the NPs. In the case of the PEO-*b*-Phis hybrid copolymer, the efficacy against all cancer cell lines was even better than free DOX. This shows that the presence of PHis favors the efficient accumulation of the drugs within the cancer cells through the rupturing of the endosomes by the “proton sponge mechanism”, improving the delivery of the drug within the cells. These results are very encouraging for these materials to be used as drug delivery carriers for anticancer agents. 

## 4. Conclusions

In this work, three series of novel hybrid amphiphilic terpolymers have been synthesized from the general type *m*PEO-*b*-P(Cys)-*b*-P(His) exhibiting different PCys topology, i.e., either between the PEO and PHis blocks, at the end of the polymeric chain or randomly distributed along the PHis chain. The terpolymers self-assemble to afford empty NPs mainly exhibiting the core–shell micellar structured NPs with multivesicular core. The polymeric materials can encapsulate the anticancer drug DOX to result in NPs exhibiting pH and redox responsiveness due to the PHis and PCys moieties, respectively, while PEO is always at the outer periphery presenting “stealth” properties, as z-potential measurements revealed. The encapsulated DOX was released in a controlled manner upon both stimuli, pH and GSH concentration. Depending on the PCys topology, NPs with different structures as well as release profiles were achieved. When the PCys is in the middle of the polymeric chain, core–shell micelles are formed, while the crosslinked PCys layer do not allow the leakage of the drug under healthy pH and GSH conditions. When the PCys is at the edge of the chain, vesicular structures are formed with gradual release of DOX depending on both stimuli. Finally, when PCys is randomly distributed, less robust core–shell micellar structures with a multivesicular structured core are formed that present a gradual release of the drug concerning both stimuli. The antiproliferative activity of these “smart” DOX-loaded NPs was tested in three breast cancer cell lines (MCF-7, T-47D and MDA-MB231) and the results revealed similar activity to DOX. Doxorubicin continues to be a cornerstone of anticancer chemotherapy being first line drug in different types of cancers and is probably the most commonly prescribed anticancer drug. However, doxorubicin suffers from severe side effects, the development of drug-induced toxicity, mainly cardiotoxicity, and the development of drug resistance being the most significant [51,52]. The cardiotoxicity is dose-dependent and the dose-limiting side effect of the drug may even result in the withdrawal of doxorubicin from the chemotherapeutic regimen. Drug resistance against doxorubicin and anthracyclines in general often occurs via the upregulation of MDR (Multi Drug Resistant) genes that control three different types of efflux proteins, pumping out of the cells, thus the drug reducing its intracellular concentration and ultimately diminishing its anticancer efficacy. To address these important clinical drawbacks of doxorubicin and because of the importance of this drug in oncology, drug delivery systems have been employed. The efforts so far have led to the development and clinical use of only two such systems: the liposomal doxorubicin and the pegylated liposomal formulation, while several other efforts have failed to advance such systems in the clinics. Thus, in this context, and if the subsequent evaluation of these “smart” systems show promise in addressing these pitfalls of the drug in vitro and most importantly in vivo in animal models of cancer, these materials could be very promising candidates in cancer treatment.

## Data Availability

Not applicable.

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
