# Peer review of "Influence of the Topology of Poly(L-Cysteine) on the Self-Assembly, Encapsulation and Release Profile of Doxorubicin on Dual-Responsive Hybrid Polypeptides"

_pharmaceutics, 2023, doi:10.3390/pharmaceutics15030790_

Round 1

Reviewer 1 Report

The authors presented a series of hybrid block copolypeptides that showed versatile self-assembly behavior based on tuning the topology of poly(L-Cysteine). Furthermore, the encapsulation and release profile of Dox were studied. Particularly, stimuli-responsive drug release can be achieved based on stimuli-responsive polypeptide segments.

1. Mulitivesicular structures are actually compound micelle, not vesicle. Please search the references.

2. Membrane permeability could be studied. This is key point of vesicle structure for tuning drug release, that also might be used for enzyme delivery (as nanoreactor, https://doi.org/10.1002/anie.202004180).

3. I suggested the authors should refine the introduction with improving the function of drug delivery a overall philosophy based on your polypeptide design (https://doi.org/10.1021/jacs.0c09029).

4. In vitro activity of nanomedicine is usually less than free drug.

5. I believe this work might be more suitable for Polymers.

Author Response

Manuscript ID: pharmaceutics-2202834

We would like to thank the reviewer for the constructive comments that really improved our manuscript.

Please find below the point-to-point answers to the comments of Reviewer #1:

The authors presented a series of hybrid block copolypeptides that showed versatile self-assembly behavior based on tuning the topology of poly(L-Cysteine). Furthermore, the encapsulation and release profile of Dox were studied. Particularly, stimuli-responsive drug release can be achieved based on stimuli-responsive polypeptide segments.

  1. Mulitivesicular structures are actually compound micelle, not vesicle. Please search the references.

- We thank the reviewer for his comment, we made appropriate corrections on the manuscript on pages 14, 16, 17, 20, 23 and 24.

  1. Membrane permeability could be studied. This is key point of vesicle structure for tuning drug release, that also might be used for enzyme delivery (as nanoreactor, https://doi.org/10.1002/anie.202004180).

- We would like to thank the reviewer for his suggestion. We will make it in a forthcoming paper.

  1. I suggested the authors should refine the introduction with improving the function of drug delivery a overall philosophy based on your polypeptide design (https://doi.org/10.1021/jacs.0c09029).

We would like to thank the reviewer for his comment. Extended corrections and additions were performed at the Introduction, on page 2.

  1. In vitro activity of nanomedicine is usually less than free drug.

- We would like to thank the reviewer for his comment. We mentioned and discussed this on Page 23. Lines 931-934.

  1. I believe this work might be more suitable for Polymers.

We would like to thank the reviewer for his comment. We believe that since the polymeric materials used are composed of biocompatible and biodegradable materials that showed no toxicity after the in vitro tests at various cell lines it can be published on a Bio-related journal such as PHARMACEUTICS. This is supported by the extended in vitro tests with various cell lined performed that showed a very good delivery of the drugs as compared to other nanoparticulate systems due to the combination of the crosslinked nature and functional materials that are composed.

Reviewer 2 Report

This manuscript described a complex delivery system composed of poly (ethylene oxide), poly(L-cysteine), and poly (histidine) that is sensitive to pH and reduction by GSH. The chemistry and characterization of materials are extensive. The release studies and CD studies provided more information about the drug delivery and secondary structure properties. The work is interesting and provides insights into designing a new generation of drug delivery agents composed of polypeptides along with polymers that could have some targeting properties. However, the new polymers did not improve the efficacy of Dox in cancer cells. The activity should have been tested in Dox-resistance and heart cells.  There are a few items that need to be considered before accepting this manuscript:

1.       Scheme 1 (supposed to be Figure 1) is confusing.  Helical structures s shown for PHIs and PCys. Instead, I suggest using square blocks for each component and labeling each block in each polymer.  The legend should explain wach polymer in more detail.

2.       Materials and Methods: Materials need to be moved from the Supplementary information to here.

3.       The authors need to explain better how they control the polymerization and size of polymer peptides. MALDI mass spectroscopy along with SEC should provide the range, but no mass data were provided.

4.       Define micelle and vesicle when describing the differences between nanoparticles.

5.       Cytotoxicity data do not make sense. There is a negative growth rate, especially in Figures 5a and 5b. Even considering the standard deviations, these are out of range.

6.       The cytotoxicity needs to be tested against normal breast cells and heart cells to see selectivity toward cancer cells if any.  The major issue with Dox is cardiotoxicity and not delivery.

7.       Another major issue with Dox is resistance. The activity needs to be tested against Dox-resistant cells to show the advantage of the polymers.

8.       Figures need to be labeled more properly rather than repeated text. See Figure Legend 5, use A, B, and C rather than writing the same words.

9.       Add some references about Dox resistance and cardiotoxicity and what has been done to circumvent it using peptides.

Author Response

Manuscript ID: pharmaceutics-2202834

We would like to thank the reviewer for the constructive comments that really improved our manuscript.

Please find below the point-to-point answers to the comments of Reviewer #2:

This manuscript described a complex delivery system composed of poly (ethylene oxide), poly(L-cysteine), and poly (histidine) that is sensitive to pH and reduction by GSH. The chemistry and characterization of materials are extensive. The release studies and CD studies provided more information about the drug delivery and secondary structure properties. The work is interesting and provides insights into designing a new generation of drug delivery agents composed of polypeptides along with polymers that could have some targeting properties. However, the new polymers did not improve the efficacy of Dox in cancer cells. The activity should have been tested in Dox-resistance and heart cells.  There are a few items that need to be considered before accepting this manuscript:

  1. Scheme 1 (supposed to be Figure 1) is confusing.  Helical structures s shown for PHIs and PCys. Instead, I suggest using square blocks for each component and labeling each block in each polymer.  The legend should explain wach polymer in more detail.

- We thank the reviewer for his comment, appropriate additions were performed to clarify the structure of the polymers at the legend of Scheme 1.

  1. Materials and Methods: Materials need to be moved from the Supplementary information to here.

-We thank the reviewer for his comment, we moved the section “Materials” from the supplementary information to the main text of the article on page 4.

  1. The authors need to explain better how they control the polymerization and size of polymer peptides. MALDI mass spectroscopy along with SEC should provide the range, but no mass data were provided.

- We thank the reviewer for his comment. MALDI could not be performed properly due to the crosslinked nature of the polymers that occurs in a small amount even before the crosslinking with H2O2. So we relied on the results from SEC which are in very good agreement to the stoichiometric one. The conditions under which is performed the synthesis of the polymers, ensure their high degree of molecular and compositional homogeneity.

  1. Define micelle and vesicle when describing the differences between nanoparticles.

-We thank the reviewer for his comment. Appropriate corrections were performed on the manuscript, since this was also the comment of Reviewer #1.

  1. Cytotoxicity data do not make sense. There is a negative growth rate, especially in Figures 5a and 5b. Even considering the standard deviations, these are out of range.

-We apologize for missing the explanation of this observation and any inconvenience due to this. Based on the formula for the calculation of the growth rate (see paragraph 2.7 under materials and methods) negative values denote cytotoxic activity as this means that the treated population ends up with less cells and thus less O.D. as compared to the starting population. We now have added an explanation for the negative values in the figure legend of Fig 5: “Negative values of growth rate denote cytotoxic activity (see paragraph 2.7 under materials and methods for the calculation of the growth rate).”

  1. The cytotoxicity needs to be tested against normal breast cells and heart cells to see selectivity toward cancer cells if any.  The major issue with Dox is cardiotoxicity and not delivery.

-We appreciate reviewer’s concern and we thank him. However, this work focuses mainly on the synthesis of hybrid polypeptide copolymers composed of PEO, PCys and Phis and aims to demonstrate that the anticancer drug DOX can effectively be loaded in differently structured NPs compared to the empty one and that copolymers can release DOX in a controlled manner, in response to pH and redox variations. We plan though to continue this work and to study the efficacy of these formulations side-by-side to the free drug and of course the safety mainly in vivo using animal models of cancer such as conventional and/or patient derived xenografts developed in our laboratories. The in vitro studies of the efficacy of the NPs in breast cancer cell lines shown in this manuscript are only preliminary to prove that the DOX-loaded NPs could be efficiently loaded and subsequently release the drug being thus potentially useful for cancer treatment.

  1. Another major issue with Dox is resistance. The activity needs to be tested against Dox-resistant cells to show the advantage of the polymers.

-We thank the reviewer for his comment. Again, reviewer’s point is good, and we appreciate it. As though discussed above the efficacy (as well as the safety) of these nanoformulations and the comparison with the free drug to see if these formulations may overcome known for doxorubicin problems are the subject of a subsequent ongoing study.

  1. Figures need to be labeled more properly rather than repeated text. See Figure Legend 5, use A, B, and C rather than writing the same words.

-We thank the reviewer. We added (a) (b) and (c) and reduced the legend accordingly.

  1. Add some references about Dox resistance and cardiotoxicity and what has been done to circumvent it using peptides.

-We thank the reviewer for his comment. To address reviewer’s concern we have now added in discussion the following part:

Doxorubicin continues to be a cornerstone of anticancer chemotherapy being first line drug in different types of cancers and most probably is the most commonly prescribed anticancer drug. However, doxorubicin suffers from severe side effects, the development of drug-induced toxicity, mainly cardiotoxicity, and the development of drug resistance being the most significant [https://doi.org/10.1177/1078155219877931;https://doi.org/10.1016/j.biopha.2017.09.059]. The cardiotoxicity is dose-dependent and the dose-limiting side effect of the drug which may even result to the withdrawal of doxorubicin from the chemotherapeutic regimen. Drug resistance against doxorubicin and anthracyclines in general occurs often via the upregulation of MDR (Mutli Drug Resistant) genes that control three different types of efflux proteins, pumping out of the cells thus the drug, reducing its intracellular concentration and ultimately diminishing its anticancer efficacy. To address these important clinical drawbacks of doxorubicin and because of the importance of this drug in oncology drug delivery systems have been employed. The efforts so far led to the development and clinical use of only two such systems the liposomal doxorubicin and the pegylated liposomal formulation while several other efforts failed to advance such systems in the clinics. Thus, in this context, and if the subsequent evaluation of these “smart” systems show promise in addressing these pitfalls of the drug in vitro and most importantly in vivo in animal models of cancer these materials could be very promising candidates in cancer treatment.

Reviewer 3 Report

There is no novelty in the work. There are lot of reports with dual responsive based release for doxorubicin and other drugs.

Author Response

Manuscript ID: pharmaceutics-2202834

Please find below the point-to-point answers to the comments of Reviewer #3:

  1. There is no novelty in the work. There are lot of reports with dual responsive based release for doxorubicin and other drugs.

We thank the reviewer for his comment.  On this work, we presented the influence of the topology of poly(L-cystein) along the polypeptidic chain of poly(L-histidine) on a block terpolymers containing also poly(ethylene oxide). The polymeric materials were completely novel. In addition, to our knowledge, there is no other work published to study the the influence of the topology of poly(L-cystein) on the self-assembly without and with encapsulated drug Doxorubicin. The results showed that the topology played a crucial role on the release profile of the drug, since it influenced the self-assembly of the materials.

In most works that have been published so far, the nanoformulations and the delivery of drugs through a nanoparticulate system show much lower activity on the treatment of cancer cell lines concerning the growth curves as compared to the growth by the treatment with free drug, as one of the reviewer also indicated.   As a consequence of the functional materials used in this work, the growth rate curves against three different cancer cell lines showed similar or even better results as compared to the free drug and much better results to the commercially available chemotherapeutic used DOXIL. This shows that the nanoparticles formed and studied are very promising and will be very interesting to Materials as well as Pharmaceutical scientists.

Round 2

Reviewer 2 Report

The authors responded to most of my comments although some experiments are planned for the future. 

Reviewer 3 Report

Manuscript can be accepted with minor correction of typo and improvement in quality of picture.